# Tissue Factor Pathway Inhibitors as Potential Targets for Understanding the Pathophysiology of Preeclampsia

**DOI:** 10.3390/biomedicines11051237

**Published:** 2023-04-22

**Authors:** Hiroshi Kobayashi, Sho Matsubara, Chiharu Yoshimoto, Hiroshi Shigetomi, Shogo Imanaka

**Affiliations:** 1Department of Gynecology and Reproductive Medicine, Ms.Clinic MayOne, 871-1 Shijo-cho, Kashihara 634-0813, Japan; shogo_0723@naramed-u.ac.jp; 2Department of Obstetrics and Gynecology, Nara Medical University, 840 Shijo-cho, Kashihara 634-8522, Japan; s.matsubara@kei-oushin.jp (S.M.); chiharu-y@naramed-u.ac.jp (C.Y.); hshige35@gmail.com (H.S.); 3Department of Medicine, Kei Oushin Clinic, 5-2-6 Naruo-cho, Nishinomiya 663-8184, Japan; 4Department of Obstetrics and Gynecology, Nara Prefecture General Medical Center, 2-897-5 Shichijyonishi-machi, Nara 630-8581, Japan; 5Department of Gynecology and Reproductive Medicine, Aska Ladies Clinic, 3-3-17 Kitatomigaoka-cho, Nara 634-0001, Japan

**Keywords:** coagulation, fibrinolysis, preeclampsia, tissue factor, tissue factor pathway inhibitor

## Abstract

Background: Preeclampsia is a hypertensive disorder of pregnancy that causes maternal and perinatal morbidity and mortality worldwide. Preeclampsia is associated with complex abnormalities of the coagulation and fibrinolytic system. Tissue factor (TF) is involved in the hemostatic system during pregnancy, while the Tissue Factor Pathway Inhibitor (TFPI) is a major physiological inhibitor of the TF-initiated coagulation cascade. The imbalance in hemostatic mechanisms may lead to a hypercoagulable state, but prior research has not comprehensively investigated the roles of TFPI1 and TFPI2 in preeclamptic patients. In this review, we summarize our current understanding of the biological functions of TFPI1 and TFPI2 and discuss future directions in preeclampsia research. Methods: A literature search was performed from inception to 30 June 2022 in the PubMed and Google Scholar databases. Results: TFPI1 and TFPI2 are homologues with different protease inhibitory activities in the coagulation and fibrinolysis system. TFPI1 is an essential physiological inhibitor of the TF-initiated extrinsic pathway of coagulation. On the other hand, TFPI2 inhibits plasmin-mediated fibrinolysis and exerts antifibrinolytic activity. It also inhibits plasmin-mediated inactivation of clotting factors and maintains a hypercoagulable state. Furthermore, in contrast to TFPI1, TFPI2 suppresses trophoblast cell proliferation and invasion and promotes cell apoptosis. TFPI1 and TFPI2 may play important roles in regulating the coagulation and fibrinolytic system and trophoblast invasion to establish and maintain successful pregnancies. Concentrations of TF, TFPI1, and TFPI2 in maternal blood and placental tissue are significantly altered in preeclamptic women compared to normal pregnancies. Conclusions: TFPI protein family may affect both the anticoagulant (i.e., TFPI1) and antifibrinolytic/procoagulant (i.e., TFPI2) systems. TFPI1 and TFPI2 may function as new predictive biomarkers for preeclampsia and navigate precision therapy.

## 1. Introduction

Preeclampsia is a pregnancy-specific multisystem disorder characterized by hypertension and proteinuria at 20 weeks of gestation, a leading cause of perinatal mortality and morbidity worldwide [1,2]. Preeclampsia is associated with shallow invasion of trophoblast cells, which involves the dysfunction of vascular endothelial cells and the imbalance between angiogenic and antiangiogenic factors, resulting in life-threatening thrombosis [3,4,5]. The cause of thrombotic complications is multifactorial, resulting from an imbalance between the coagulation and fibrinolytic systems [6]. Tissue Factor (TF) is a major initiator of the extrinsic coagulation pathway and is involved in the hemostatic system during pregnancy and delivery [7,8]. TF has been reported to contribute to the generation of hypercoagulable and prothrombotic states under physiological conditions (e.g., normal pregnancy) and pathological conditions (e.g., preeclampsia and cancer) [9]. On the other hand, the Tissue Factor Pathway Inhibitor (TFPI) has been reported to be a major physiological inhibitor of the TF-dependent initiation phase of the coagulation pathway [10]. Indeed, in cancer patients, upregulation of TF and downregulation of TFPI were initially shown to increase the probability of developing hypercoagulability and thrombotic complications [11]. TFPI1 and TFPI2 are homologous proteins that are encoded by separate genes [12,13]. Furthermore, two TFPI1 isoforms, TFPIα and TFPIβ, are generated by alternative mRNA splicing [14]. Recent studies have shown that decreased TFPI1 [11] levels or increased TFPI2 [15,16] levels are associated with an increased risk of thromboembolism in cancer patients, suggesting that TFPI1 and TFPI2 may exhibit different biological activities. On the other hand, a number of studies have shown that not only TF, but also TFPI1 and TFPI2 are elevated, to varying degrees, in preeclamptic patients [17,18,19,20]. Trophoblasts predominantly express both TF and TFPIs to control placental hemostasis [21,22]. Furthermore, in addition to its role in the coagulation and fibrinolysis system, TFPI1 and TFPI2 have also been found to regulate the proliferation, invasion, differentiation, and apoptosis of cancer cells and trophoblast cells [3,19,23]. However, the mechanisms underlying the biological functions of TFPI1 and TFPI2 remain elusive. To date, no publications have comprehensively investigated the roles of plasma and placental TFPI1 and TFPI2 in preeclamptic patients. The aim of this review is to summarize recent data on the role of TFPI1 and TFPI2 in normal pregnancy and preeclampsia and discuss the impacts of TFPI protein family on the pathophysiology of preeclampsia.

## 2. Materials and Methods

### Search Strategy and Selection Criteria

A literature search was conducted in PubMed and Google Scholar to identify relevant studies published up to 30 June 2022, using the keywords: tissue factor pathway inhibitor, tissue factor, preeclampsia, coagulation, and fibrinolysis. The search terms were combined with Boolean operators AND and OR, as described in Table 1. Included studies were the publication of original studies in the English language and reference lists in review articles. Duplicated studies, literature unrelated to the research topic, non-English publications, letters to the editor, and poster presentations were excluded. The first identification phase included electric database searched, hand searched, and reference list of collected article and review article to identify additional relevant articles (Figure 1). Duplicates were removed during the second screening phase. Titles and abstracts were read to remove inappropriate papers. The final eligibility phase included the full-text articles for analysis after excluding inappropriate articles. Articles that did not focus on preeclampsia (e.g., inflammation, sepsis, and atherosclerosis) were excluded as insufficient data. Articles that could not distinguish between TFPI1 and TFPI2 were also excluded such as when only the term “TFPI” was used in the text.

## 3. Results

### 3.1. Selection of Studies

The initial searches yielded 942 citations. Screening of title and abstract review combined with removal of duplicates resulted in 231 articles for full-text evaluation. Twenty-nine articles were selected for this review after removal of inappropriate articles (Figure 1).

This figure shows flow diagram of selection process.

### 3.2. The Coagulation and Fibrinolytic System in Normal Pregnancy

Pregnancy is associated with local and systemic hypercoagulability and hypofibrinolysis [24]. Factors that change during pregnancy are an increase in TF and a number of clotting factors (Factor I (FI, fibrinogen), FII (prothrombin), FVII, FVIII, FIX, FX, FXII, and von Willebrand factor), a decrease in protein S and activated protein C (APC) levels, and inhibition of fibrinolysis (e.g., plasminogen activator inhibitor (PAI)-1 and PAI-2) [23]. Many of the clotting factors, except FXI, increase during pregnancy [24]. Plasma concentrations of these clotting factors increase during pregnancy, reaching peak levels around term [6,21]. Not only do extravillous trophoblasts (EVT) invade the decidualized endometrium, but also decidual cells express two key modulators of hemostasis, TF and PAI-1, to maintain hemostasis during placentation [25,26]. In addition, decidualized cells downregulate the expression of plasminogen activators (PAs), matrix metalloproteinases (MMPs), and the vasoconstrictor, endothelin-1 [26]. Therefore, decidualized endometrium alters the hemostatic and fibrinolytic capacity and extracellular matrix turnover to prevent hemorrhage during vascular remodeling [26]. This section reviews the physiological changes in the coagulation and fibrinolytic system during normal pregnancy, focusing on TF and TFPI as markers of hemostatic balance.

#### 3.2.1. Tissue Factor (TF)

The cell-surface glycoprotein, TF, is a trigger to initiate the extrinsic coagulation cascade and activates the hemostatic system through the binding to FVII/FVIIa [7,27]. The resulting TF/FVIIa complex then activates FIX and FX by limited proteolysis, leading to the generation of thrombin. FXa is a superior activator of TF/FVII, because the interaction of FXa with TF/FVIIa is stronger than FIXa [28]. Therefore, TF confers the hypercoagulable and prothrombotic state in pregnant women. Indeed, TF expression in maternal plasma and placental tissue increases to control bleeding during pregnancy and at delivery [7]. Maternal plasma TF concentration increases 1 h after delivery and decreases at day 1 postpartum [7].

#### 3.2.2. Tissue Factor Pathway Inhibitor (TFPI)

##### Domain Architecture

TFPI1 and TFPI2 are members of the Kunitz-type serine proteinase inhibitor family [12,13] (Figure 2). TFPI1 is a 42 kDa serine proteinase inhibitor consisting of a negatively charged N-terminus, three tandem Kunitz-type (K1, K2, and K3) domains, and a positively charged C-terminus [29,30]. Furthermore, *TFPI1* gene produces a structurally and functionally distinct isoform, *TFPIβ* [29]. Unlike TFPI1α, TFPI1β contains two Kunitz-type domains (K1 and K2) and a specific C-terminal glycosylphosphatidylinositol (GPI)-attachment signal peptide TFPIβ is anchored to the cell membrane via the GPI-anchor attachment peptide. TFPI1 exists in two forms, the secreted soluble form and the membrane-anchored form, where both have multiple functions [31]. In addition, TFPI2, also known as placental protein 5 (PP5) or matrix serine protease inhibitor (MSPI), is a 32 kDa serine proteinase inhibitor consists of three Kunitz-type domains (K1, K2, and K3) connected to short N-terminus and basic C-terminus [29,30]. TFPI1 is often described as TFPI or TFPIα, but in some publications, TFPI may include both TFPI1 and TFPI2. Therefore, caution should be taken in the interpretation of TFPI result.

This figure shows the domain architecture and their functions of TFPI1 (TFPIα and TFPIβ) and TFPI2.

TFPI1 inhibits the TF-initiated coagulation cascade and thrombin generation via inactivating the TF-FVIIa and prothrombinase (FXa-FVa) complexes [31,32]. The K1 and K2 domains of TFPI1 inhibit the FVIIa and FXa activation, respectively. The K3 domain and C-terminus are required for protein S and FV/FVa binding, respectively. Protein S is a potent cofactor for APC in regulating the intrinsic coagulation pathways and enhances the TFPI action. In addition, TFPIα interacts with the extracellular matrix-related proteins, including thrombospondin 1 (THBS1) [33], syndecan 4 (SDC4) [34], and glypican 3 (GPC3) through its C-terminus [34]. On the other hand, unlike TFPI1, the K1 domain of TFPI2 inhibits plasmin, plasma kallikrein, FXIa, trypsin, chymotrypsin, and cathepsin G, very weakly inhibits FVIIa, but not urokinase-type plasminogen activator, tissue-type plasminogen activator, or thrombin [35]. The K2 domain does not inhibit FXa activity, but instead binds the globular C1q receptor (gC1qR) [36]. The gC1qR interacts with a variety of serum components and extracellular matrix components, including C1q, TFPI2 (estimated Kd: approximately 70 nM), high-molecular weight kininogen, FXII, vitronectin, and hyaluronic acid [37,38]; however, the biological function of gC1qR on TFPI2 is still unknown.

##### Sources

Vascular endothelial cells, platelets, and trophoblasts are major sources of TFPIα, TFPIβ, and TFPI2 [32], but the platelet α-granules do not contain TFPIβ [39]. Platelet activation actually triggers the release of TFPI1 and TFPI2. Platelet-derived TFPI1 downregulates the initiation phase of coagulation, whereas TFPI2 not only regulates the coagulation pathway, but also promotes fibrinolysis [39]. TFPI1 and TFPI2 are produced in the placenta (i.e., the extravillous trophoblast (EVT) column, endovascular invasive trophoblast cells, trophoblast giant cells, syncytiotrophoblast of villous structures) and embryonic tissue [23,40]. Since endovascular trophoblasts replace the endothelium for spiral artery remodeling at the site of placentation, TFPI localizes to the surface of the blood-contacting lumen. TFPI2 in particular is highly abundant in villous cytotrophoblasts and syncytiotrophoblasts through the binding to membrane-anchored proteoglycans (e.g., glypican or syndecan) [13,19,32,34]. Some released TFPI1 and TFPI2 enter the maternal circulation. Therefore, TFPI1 and TFPI2 modulate activation of blood coagulation cascades and counteract prothrombotic challenges triggered by placentation.

##### The Functional Diversity of TFPI1

Total TFPI concentrations in maternal plasma increase during the first half of pregnancy, remain constant during the second half [41], and decrease after labor [32,42]. Anticoagulant properties of TFPI1 have been demonstrated in several preclinical and clinical studies. In vivo TFPI1 deficiency in knockout mice results in a severe and uncontrolled coagulation activation [13]. In vivo TFPI deficiency may lead to inadequate remodeling of the spiral arteries due to the failure of invasive trophoblasts through excessive fibrin accumulation. In mice, low levels of TFPI1 are likely to be associated with an increased risk of thrombosis [10]. The lack of TFPI K1 domain also leads to elevated thrombin-antithrombin (TAT) levels, aberrant renal fibrin deposition, extensive brain ischemia and infarction, and an increased risk of the TF-induced pulmonary thromboembolism [43]. In addition, decreased TFPI activity may be a useful predictive biomarker of the thrombotic events in women using oral contraceptives [44]. Overall, TFPI1 functions as a regulator to control the TF-induced hypercoagulability (Table 2).

Furthermore, in addition to controlling coagulation, TFPI1 has been reported to be involved in EVT cell invasion [23]. EVT cells resemble malignant tumor cells in terms of proliferation and invasion, and the biological functions and underlying molecular mechanisms of TFPI1 in cancer cells have been extensively studied. Therefore, we first summarize the current knowledge of TFPI1 in cancer cells and then discuss its fundamental characteristics in EVT cells. Emerging evidence suggests that TFPI1 inhibits cancer cell proliferation and invasionTFPI1 has been reported to negatively regulates tumor cell proliferation and invasion through suppression of the TF-dependent protease-activated receptor 2 (PAR2) activation [49,50]. PAR2 is one of the four members of the G protein-coupled receptor family (PAR1-4) [51]. PAR2 facilitates tumor aggressiveness through mitogen-activated protein kinase, nuclear factor-kappa B, protein kinase C (PKC)α, or extracellular signal-related kinase signaling [50,51,52]. Moreover, TFPI inhibited breast cancer cell proliferation and invasion by downregulating the extracellular signal-regulated kinase (ERK)/p38 mitogen-activated protein kinases (MAPK) signaling pathway [53]. Additionally, plasma TFPI1 levels in patients with non-small cell lung cancer (NSCLC) and breast cancer were significantly lower in the VTE group than in the non-VTE group [11,53]. Furthermore, NSCLC patients with metastasis had significantly lower TFPI1 levels than those without metastasis [11]. These data indicated that downregulation of TFPI1 could predict DVT and poor prognosis in NSCLC patients. Overall, TFPI1 not only plays a fundamental role in coagulation, but also shows anti-proliferative and anti-invasive properties in several cancers. On the other hand, TFPI1 induces trophoblast stem cell differentiation to EVT cells and knockdown of TFPI1 restricts EVT cell invasion [23], suggesting that TFPI1 is involved in EVT cell invasion and placentation. TFPI1 significantly attenuated the proliferation and invasion of cancer cells, whereas it exerts the opposite effect in trophoblast cells. It is currently unclear why cancer cells and differentiated EVT cells behave differently in response to TFPI1. Finally, a wide range of biological activities is also exerted through various binding partners for TFPIα, including THBS1, SDC4, and GPC3 [54]. These extracellular matrix proteins and proteoglycans are thought to play a potential role in anticoagulant processes, trophoblast outgrowth, placental development, mammalian embryogenesis, and fetomaternal communication [54]. Collectively, TFPI1 is not only a potent endogenous inhibitor of the TF-initiated coagulation cascade, but also the regulator of trophoblast and cancer cell proliferation, migration, invasion, and differentiation.

##### The Functional Diversity of TFPI2

This subsection summarizes the available literature regarding plasma TFPI2 levels in healthy women throughout the course of pregnancy. During pregnancy, the maternal plasma concentration of TFPI2 continues to rise steadily, reaches a plateau at 36–39 weeks (282.6 ± 17.1 ng/mL, approximately 9 nM) and dramatically decreases to nearly a non-pregnant level following delivery (16.0 ± 3.6 ng/mL, approximately 0.5 nM) [32,45,55]. However, plasma TFPI2 levels during pregnancy varied widely in the literature. For example, TFPI2 levels ranged from 0.3 nM to 10 nM (10–320 ng/mL) in the third trimester [20,39,56,57]. Previous study showed that when plasma and serum TFPI2 levels were measured for the same subject, there was no significant difference between them [58], indicating that TFPI2 concentrations that differ between publications are not due to differences between serum and plasma samples. The TFPI2 concentration in maternal plasma was significantly higher than that in umbilical plasma and decreased dramatically postpartum [45], suggesting a placental origin for TFPI2. Significant correlations between placental hemostatic and angiogenic parameters have been reported in the full-term human placentas [59]. Indeed, the TFPI2 levels correlated well with the vascular endothelial growth factor (VEGF) levels. Therefore, TFPI2 is abundantly expressed at the placenta over the course of a pregnancy to facilitate blood flow via pro-angiogenesis (Table 2).

Second, recent progress has been made toward elucidating the role of TFPI2 in the regulation of the blood coagulation and fibrinolytic systems. Normal hemostasis is maintained by a complex, active, and controlled balance between pro-coagulation, anticoagulation, and fibrinolysis. Plasmin is the major effector enzyme involved in fibrinolysis and its physiological target is fibrin [60]. The K1 domain of TFPI2 exerts antifibrinolytic activity through inhibition of plasmin activity, maintaining the hypercoagulable state [35]. Plasmin also directly inactivates several clotting factors (e.g., FV, FVIII, FIX, and FX) [24,60]. TFPI2 can maintain and regulate blood coagulation by inhibiting and stabilizing the inactivation of plasmin-dependent coagulation factors. Therefore, considering the physiological functions of TFPI2, it may act as an endogenous antifibrinolytic and clotting factor stabilizer rather than an anticoagulant to control bleeding.

Third, there is clinical evidence that TFPI2 contributes to promoting the thrombus formation. Indeed, in patients with ovarian cancer, elevated serum TFPI2 levels were reported to be associated with an increased risk of VTE [15,16]. A multivariate analysis showed that a cut-off value of 12.5 pM for TFPI2 was an independent risk factor for VTE. Ovarian cancers produce large amounts of TFPI2 [61], so its concentrations in cancer tissues are expected to be much higher than those in peripheral blood. It is still unclear whether TFPI2 is associated with increased risk of development of VTE in other types of cancer as well.

Finally, TFPI2 may participate in biological processes that modulate the trophoblast survival. Unlike TFPI1, TFPI2 inhibits trophoblast cell proliferation and invasion and induces apoptosis [3,19]. In vitro experiments using TFPI2-transfected BeWo and JEG-3cell lines showed that TFPI2 inhibited the trophoblast migration, invasion and proliferation, and induced apoptosis [62]. In addition, downregulation of TFPI2 increased the trophoblast cell invasion through upregulating MMP2 and MMP9 expression [3]. Therefore, TFPI2 is thought to control placentation by regulating the trophoblast proliferation, invasion, and differentiation [31]. TFPI2 abundantly expressed by trophoblasts not only inhibits fibrinolysis and suppresses inactivation of clotting factors to control bleeding through the intrinsic coagulation system, but may also play an important role in placentation.

Overall, TFPI2 exhibits distinctly different biological activities not only in the coagulation and fibrinolytic system, but also in trophoblast cell invasion and placentation compared to TFPI1.

### 3.3. The Coagulation and Fibrinolytic System in Preeclampsia

To date, several researchers have reported the coagulation and fibrinolytic changes in women with preeclampsia. Preeclampsia is associated with the increased t-PA, thrombomodulin, TAT, PAI-1, TF, TFPI1, and TFPI2 levels, decreased fibrinogen, antithrombin III, and PAI-2 levels, and increased fibrinolysis (e.g., D-dimer) compared to the normotensive controls [63,64,65,66]. Overall, there is evidence for increased intravascular coagulation, extravascular fibrin deposition, and fibrin turnover in women with preeclampsia, which is potentiated by changes in endogenous coagulation and fibrinolytic regulatory systems [64]. Here, we summarize the biological activities of TFPI1 and TFPI2 in preeclampsia (Table 2). To date, however, no articles have compared total TFPI, free TFPI1, and free TFPI2 in plasma, platelets, and placental tissues of women with normal pregnancy and preeclampsia. For example, an immunoassay from American Diagnostica [17] detects both TFPI1 and TFPI2, so it measures total TFPI concentration.

#### 3.3.1. The Role of TFPI1

The maternal TF concentration was significantly higher in patients with preeclampsia than that of women with a normal pregnancy (median, 1187 pg/mL vs. 291.5 pg/mL) [17] (Figure 3). Furthermore, the maternal TFPI1 concentration was higher among patients with preeclampsia than in those with a normal pregnancy (median 42.3 ng/mL vs. 30 ng/mL) [18]. Another study also showed that TFPI concentration was significantly higher in samples from patients with preeclampsia compared with normal pregnancies (median, 87.5 ng/mL vs. 66.1 ng/mL) [17]. In general, women with preeclampsia have been shown to have elevated levels of TF and TFPI1 compared to those with normal pregnancies. In particular, maternal TFPI1 concentrations were found to be 50- to 200-times higher than TF concentrations [67]. Interestingly, in patients with preeclampsia, the TFPI1-to-TF ratio was significantly lower than that of normal pregnancy [17]. Therefore, the TFPI1-to-TF ratio may be more informative to predict preeclampsia, supporting that preeclampsia is characterized by the imbalance between procoagulant and anticoagulant activities [32].

TFPI2 concentrations in maternal blood and placental tissue may be dependent on GPC3 expression in preeclamptic placentas. Downregulation of glypican-3 expression in placenta results in elevated circulating levels of TFPI2.

#### 3.3.2. The Role of TFPI2

Next, we summarize the TFPI2 concentrations in maternal blood and placental tissue from healthy pregnant women and preeclamptic patients (Figure 3). Maternal serum TFPI2 concentrations have been reported to be significantly increased in the preeclampsia group relative to the normal pregnant group [19]. In particular, the altered expression of TFPI2 was reportedly identified in early forms of preeclampsia [19]. In contrast, other studies demonstrated that maternal plasma TFPI2 concentrations were significantly lower in preeclamptic pregnant women compared with normal healthy pregnant women [9,45]. The results were inconsistent throughout the literature regarding whether maternal circulating TFPI2 is increased in preeclamptic patients.

Local hemostasis at the trophoblast level in normal placenta is characterized by increased TF expression, low expression of TFPI1 [21], and increased TFPI2 expression [22] (Figure 3, left). TF overexpression in the placenta of preeclamptic women promotes the hypercoagulable state [9]. Additionally, TFPI2 is found to be overexpressed in human preeclamptic placenta compared with control placenta [19,45]. The placenta is the main site responsible for TFPI2 synthesis, with a release of TFPI2 into the maternal circulation [51]. Placental TFPI2 expression was negatively correlated with placental weight and birth weight [19], suggesting that it suppresses placental formation. Interestingly, experimental animal models have demonstrated that inhibition of the placental TFPI2 expression may lead to prevention of the development of preeclampsia [20]. On the other hand, TFPI2 expression has also been reported to be significantly lower in the preeclampsia group than in controls [9,20]. Differences in TFPI2 expression in preeclamptic patients have been reported to depend on the degree of expression of the TFPI2-binding protein, glypican-3, in placental tissue [20]. In fact, in vitro experiments using the TFPI2 transfected cell lines demonstrated that GPC3 could anchor TFPI2 to the placental trophoblast cell surface [20]. Therefore, GPC3 deficiency induces the release of TFPI2 into the maternal circulation [20], suggesting that maternal circulating TFPI2 levels depends on the expression of glypican-3 in the preeclamptic placenta (Figure 3, middle and right). Therefore, the inconsistent results may be due to differential expression of placental glypican-3.

## 4. Discussion

Recent studies have focused on TF and TFPI protein family as the placenta-specific regulators of the hemostatic balance in preeclampsia. This review aims to provide a better understanding of TFPI1 and TFPI2 in the pathophysiology of preeclampsia and discuss the underlying mechanisms.

First, concentrations of TF and TFPI1 are significantly increased in maternal plasma of preeclamptic women compared with normal pregnancy, while various studies have yielded inconsistent results regarding TFPI2 [17,19,20,45,68]. This is likely because TFPI2 concentrations in maternal plasma depend on the differential expression of placental GPC3, a TFPI2-binding proteoglycan (Figure 3), but methods to make clinical identification of individual changes in the GPC3 concentration are limited. Low expression of GPC3 in placenta of preeclampsia results in an enhanced influx of TFPI2 into the maternal circulation [20]. However, the clinical significance of the correlation between maternal circulating TFPI2 levels and placental TFPI2 and GPC3 expression is still unresolved.

Second, TFPI1 functions as an anticoagulant, but TFPI2, a structural homologue of TFPI1, does not exhibit the potent anticoagulant activity. In contrast, TFPI2 inhibits plasmin-mediated fibrin clot lysis and exerts antifibrinolytic activity [39]. TFPI2 also maintains the hypercoagulable state through inhibition of plasmin-mediated inactivation of clotting factors [60]. Therefore, TFPI2 produced by the syncytiotrophoblasts would provide antifibrinolytic activity and stabilize clotting factors during pregnancy and delivery to maintain a hypercoagulable state and inhibit bleeding. Taken together, TFPI1 and TFPI2 have opposing functions on blood coagulation. The name “TFPI” may not be appropriate for TFPI2.

Third, in the field of cancer research, it is a well-known fact that TFPI2 can serve as a tumor suppressor gene [35,69,70,71,72,73,74,75]. TFPI2 has been reported to be downregulated in several types of cancer, and loss of TFPI2 function may facilitate tumor invasion and metastasis [71]. Similar to cancer cells, several literatures suggest that aberrant expression of TFPI2 suppresses trophoblast cell proliferation and invasion and promotes cell apoptosis; however, TFPI1 has an effect opposite to TFPI2 on the trophoblast invasion [3,19,31,46,62]. TFPI2, initially characterized as an endogenous inhibitor of the TF-dependent pathway, actually exerts antifibrinolytic, anti-invasive, anti-survival, and pro-apoptotic functions and is likely involved in placentation [24,35,60]. Therefore, TFPI1 and TFPI2 may play crucial roles in regulating the coagulation and fibrinolytic system and placentation to establish and maintain successful pregnancies [17,32]. Therefore, TFPI protein family exhibits diverse biological activities.

Fourth, a clinically applicable TFPI2 ELISA kit for ovarian cancer diagnosis was launched at Tosoh Corporation in 2021 [58]. A multicenter prospective cohort study revealed that TFPI2 is a novel serum biomarker for predicting ovarian cancer [9]. Furthermore, ovarian cancer patients who experienced VTE had elevated TFPI2 levels compared to those without thromboembolic events, suggesting that TFPI2 may be a biologic marker for predicting VTE [15,16]. A better understanding of the biological functions of TFPI2 may help develop optimal therapeutic strategies not only for cancer and VTE, but also for preeclampsia. Strategies to predict, prevent, and manage preeclampsia, which can cause serious complications for mother and fetus, remain a major challenge. As plasma TFPI is increased and placental growth factor (PlGF) is decreased in women with preeclampsia, the TFPI-to-PlGF ratio may be useful as a predictive marker for early prediction of preeclampsia [18]. Therefore, it is of great interest to assess which marker best predict the risk for development of preeclampsia using the TFPI1- and TFPI2-specific immunoassays. However, whether TFPI2 concentration in maternal blood reflects placental TFPI2 expression remains an issue for the future.

Finally, there is an issue to resolve. Variability in TFPI2 measurements may be due to commercially available immunoassay kits containing various antibodies that recognize different epitopes of TFPI molecules. A proper methodology to quantify TFPI2 protein levels should be developed, standardized, and validated.

In conclusion, this review summarizes recent advances in our knowledge of TFPI1 and TFPI2 in pathogenesis of preeclampsia. TFPI protein family may have multiple roles in maintaining homeostasis of the coagulation and fibrinolytic system and regulating trophoblast cell proliferation and invasion under physiological and pathological conditions. One future direction is to identify and validate TFPI protein family as clinically useful biomarkers for predicting disease onset.

## Figures and Tables

**Figure 1 biomedicines-11-01237-f001:**
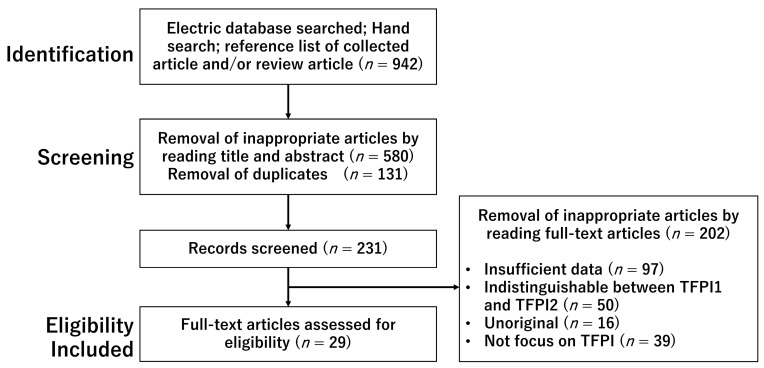
The number of articles identified by searching for keyword combinations.

**Figure 2 biomedicines-11-01237-f002:**
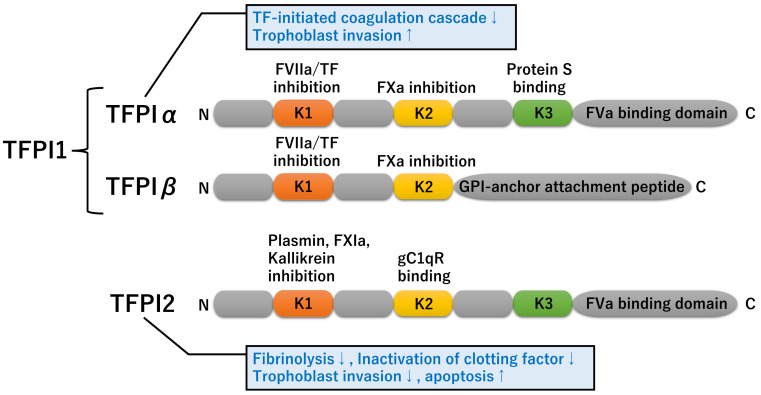
Conserved motif arrangement in proteins of the TFPI family.

**Figure 3 biomedicines-11-01237-f003:**
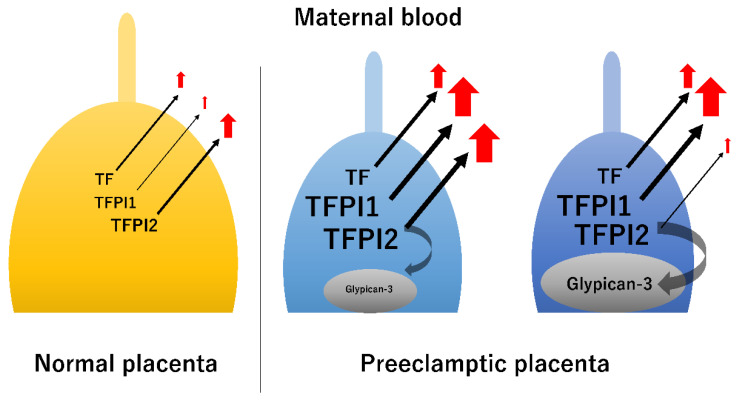
Concentrations of TF, TFPI1, and TFPI2 in maternal blood and placental tissue from healthy pregnant women and preeclamptic patients. The thicker the arrow in the figure, the greater the production volume.

**Table 1 biomedicines-11-01237-t001:** The search strategy.

Search Mode	The Keyword and Search Term Combinations
Search term 1	Tissue factor pathway inhibitor OR TFPI OR TFPI1 OR TFPI2
Search term 2	Tissue factor OR TF
Search term 3	Preeclampsia OR Eclampsia
Search term 4	Coagulation
Search term 5	Fibrinolysis
Search	Search term 1 AND Search term 2
	Search term 1 AND Search term 3
	Search term 1 AND Search term 4
	Search term 1 AND Search term 5
	Search term 2 AND Search term 3
	Search term 1 AND Search term 3 AND Search term 4
	Search term 1 AND Search term 3 AND Search term 5

**Table 2 biomedicines-11-01237-t002:** The functional diversity of TFPI1 and TFPI2.

	Effects on the Blood Coagulation and Fibrinolytic Systems	Effects on Trophoblasts					Effects on Cancer Cells	The Underlying Mechanisms	Preeclampsia		
									Maternal TFPI Concentration	Placental TFPI Concentration	The TFPI-to-TF Ratio
TFPI1	TF-initiated coagulation cascade	↑Invasion ↑Differentiation					↓Invasion ↓The risk of thrombosis	↓PAR2, MAPK, ERK, NF-kB, PKC	PE > NP [17,18]	PE > NP [18]	PE < NP [32]
TFPI2	Antifibrinolytic activity through inhibition of plasmin activity and Inactivation of several clotting factors	↓Invasion	↑Proliferation		↓Apoptosis	ref. [3]	↓Invasion ↑The risk of thrombosis	↓MMP2	PE > NP [19] or PE < NP [9,45]	PE > NP [19,45] or PE < NP [9,20]	
			↓Proliferation	↓Migration		ref. [46]					
			↓Proliferation			ref. [47]					
		↑Invasion	↑Proliferation	↑Migration	↓Apoptosis	ref. [48]					

The data in lines 1, 2, 3 and 4 of the TFPI2 section are from refs [3,46,47], and [48], respectively. PE, Preeclampsia; NP, Normal pregnancy; PAR2, protease-activated receptor 2; MAPK, mitogen-activated protein kinase; ERK, extracellular signal-regulated kinase; NF-kB, nuclear factor kappa B; PKC, protein kinase C; and MMP2, matrix metalloproteinase 2.

## Data Availability

No new data were created.

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
