# Peer review of "Tissue Factor Pathway Inhibitors as Potential Targets for Understanding the Pathophysiology of Preeclampsia"

_biomedicines, 2023, doi:10.3390/biomedicines11051237_

Round 1

Reviewer 1 Report

This is a review paper on the heterogeneity of Tissue Factor Pathway Inhibitors (TFPI)s. It focuses on the pregnant woman and the placenta. It assesses what is known about the links between the two TFPIs and placental diseases, showing the differences between TFPI-1 and TFPI-2, especially in pre-eclampsia.

This paper does not bring anything new to those who are familiar with this system, nor does it offer any original interpretation or vision, but it is a fairly good introductory didactic summary for those who are new to TFPIs in preeclampsia.

It is a pity that the authors do not include any haematologists or haemostasis specialists who are more likely to have an instinctive and sustained knowledge of TFPIs. This could have avoided some blatant approximations. For example, the authors present the tissue factor/factor VII(a) pair as activating coagulation through the activation of factor X. They seem to ignore the fact that the activation of factor VII(a) is not the same as that of factor X.  They seem to ignore the fact that coagulation activation occurs in two distinct ways. At high concentrations of TF/FVIIa, factor X is activated. But at low concentrations, this is no longer possible, as the TF/FVIIa couple must first activate factor IX (anti-haemophilic factor B), which explains why haemophiliacs bleed mainly in TF-poor organs.

I thus encourage the authors to approach professionals with a better understanding of physiological and pathological haemostasis.

Similarly, the potential didactic properties of the paper are signifcantlyn impaired by the absence of clear figures and diagrams describing the physiological role of TFPI-1 and TFPI-2 both in the haemostasis system and in trophoblastic cell physiology. The same applies to pre-eclampsia (effects of TFPIs on haemostasis and trophoblastic pathophysiology)

Furthermore, nothing specific or clear is said about TFPIs in non-pregnant women at risk of pre-eclampsia; and in non-pregnant women with a history of pre-eclampsia and/or placental disease. Nor on the effects of treatments given during pregnancy for the prophylaxis of pre-eclampsia on the TFPIs of treated women.

Furthermore, because of the predictive power of maternal circulating factors impacting on angiogenesis (sFlt1, free PlGF,...) on the occurrence of preeclampsia, we would like to know the relationships described between TFPIs, sFlt& and free PlGF.

Author Response

Answer to the reviewers

Manuscript ID: biomedicines-2322019

Type of manuscript: Review

Title: Tissue factor pathway inhibitors as potential targets for understanding the pathophysiology of preeclampsia

Authors: Hiroshi Kobayashi *, Sho Matsubara, Chiharu Yoshimoto, Hiroshi Shigetomi, Shogo Imanaka

Dear Editor in Chief:

Thank you and the reviewers for the thoughtful comments and helpful suggestions on my manuscript “Tissue factor pathway inhibitors as potential targets for understanding the pathophysiology of preeclampsia” (manuscript ID: biomedicines-2322019), authored by Hiroshi Kobayashi and colleagues. We have carefully considered each of the comments, made every effort to address the concerns raised, and applied corresponding revisions to the manuscript.

The detailed, point-by-point responses to the reviewer comments are given below, whereas the corresponding revisions are highlighted to our manuscript within the document.

We believe that our manuscript has been considerably improved as a result of these revisions, and hope that the revised manuscript is acceptable for publication in Biomedicines.

I would like to thank you once again for your consideration of our work and inviting us to submit the revised manuscript. I look forward to hearing from you.

With best regards,

Hiroshi Kobayashi, M.D., Ph.D.

Department of Gynecology and Reproductive Medicine, Ms.Clinic MayOne, Kashihara, Nara 634-0813, Japan

Department of Obstetrics and Gynecology, Nara Medical University, Kashihara, Nara 634-8522, Japan.

Tel: +81 744 29 8877

Fax: +81 744 23 6557

E-mail: hirokoba@naramed-u.ac.jp

Point-by-point responses to reviewer comments

To the Editor:

We revised the paragraphs that are highlighted in red.

Reviewer #1

Comment 1

It is a pity that the authors do not include any haematologists or haemostasis specialists who are more likely to have an instinctive and sustained knowledge of TFPIs. This could have avoided some blatant approximations. For example, the authors present the tissue factor/factor VII(a) pair as activating coagulation through the activation of factor X. They seem to ignore the fact that the activation of factor VII(a) is not the same as that of factor X.  They seem to ignore the fact that coagulation activation occurs in two distinct ways. At high concentrations of TF/FVIIa, factor X is activated. But at low concentrations, this is no longer possible, as the TF/FVIIa couple must first activate factor IX (anti-haemophilic factor B), which explains why haemophiliacs bleed mainly in TF-poor organs.

I thus encourage the authors to approach professionals with a better understanding of physiological and pathological haemostasis.

Response 1

3.2.1.         Tissue factor (TF)

We tried to be as accurate as possible. We fixed it as follows:

The resulting TF/FVIIa complex then activates FIX and FX by limited proteolysis, leading to the generation of thrombin. Compared to FIXa, FXa is a superior activator of TF/FVII, because the interaction of FXa with TF/FVIIa is stronger than FIXa [Vadivel K].

Comment 2

Similarly, the potential didactic properties of the paper are signifcantlyn impaired by the absence of clear figures and diagrams describing the physiological role of TFPI-1 and TFPI-2 both in the haemostasis system and in trophoblastic cell physiology. The same applies to pre-eclampsia (effects of TFPIs on haemostasis and trophoblastic pathophysiology)

Response 2

I agree that your comment is important. However, to date, no article compares total TFPI, free TFPI1, and free TFPI2 in plasma, platelets, and placental tissues of normal pregnant and preeclamptic women. Recently, immunoassays that specifically measure TFPI2 have emerged, but until then, most immunoassays detected both TFPI1 and TFPI2 simultaneously and measured total TFPI concentration. Also, some papers measure only total TFPI activity using a chromogenic assay. Therefore, some studies have assessed TF and TFPI in preeclamptic and normal pregnant women, but results are inconsistent.

We added Table 2 for better understanding of the reader.

Table 2. The functional diversity of TFPI1 and TFPI2

The data in lines 1, 2, 3 and 4 of the TFPI2 section are from refs 3, 46, 47, and 48, respectively.

PE, Preeclampsia; NP, Normal pregnancy; PAR2, protease-activated receptor 2; MAPK, mitogen-activated protein kinase; ERK, extracellular signal-regulated kinase; NF-kB, nuclear factor kappa B; PKC, protein kinase C; and MMP2, matrix metalloproteinase 2.

Comment 3

Furthermore, nothing specific or clear is said about TFPIs in non-pregnant women at risk of pre-eclampsia; and in non-pregnant women with a history of pre-eclampsia and/or placental disease. Nor on the effects of treatments given during pregnancy for the prophylaxis of pre-eclampsia on the TFPIs of treated women.

Response 3

Comments from the reviewers are very interesting. Di Bartolomeo et al reported in 2017 that the TFPI assay is useful for predicting placenta-mediated adverse pregnancy outcomes in high-risk women. The authors used a chromogenic assay and measured total TFPI activity, so it was not possible to measure TFPI1 and TFPI2 separately. At this time, it is inconclusive whether TFPI1 or TFPI2 are important for disease prediction and prevention, so this point was not included in our review.

Di Bartolomeo A, Chauleur C, Gris JC, Chapelle C, Noblot E, Laporte S, Raia-Barjat T. Tissue factor pathway inhibitor for prediction of placenta-mediated adverse pregnancy outcomes in high-risk women: AngioPred study. PLoS One. 2017 Mar 22;12(3):e0173596. doi: 10.1371/journal.pone.0173596.

Comment 4

Furthermore, because of the predictive power of maternal circulating factors impacting on angiogenesis (sFlt1, free PlGF,...) on the occurrence of preeclampsia, we would like to know the relationships described between TFPIs, sFlt& and free PlGF.

Response 4

Thank you for your valuable suggestion. MacDonald et al. reported that as plasma TFPI increased and placental growth factor (PlGF) decreased in women with preeclampsia, the TFPI-to-PlGF ratio may be useful as a predictive marker for early prediction of preeclampsia. Similar to the previous answer, plasma TFPI is increased in preeclamptic patients, but TFPI1 and TFPI2 could not be quantified separately. Therefore, it is not well understood which marker best predicts the outcome of preeclampsia. Moreover, a literature search with the keywords "tissue factor pathway inhibitor" and "sFlt-1" yielded one paper [3]. This study suggested that the inhibition of placental TFPI2 and HIF-1α/VEGF might be one of the potential mechanisms underlying the protective effects of Vitexin (VI) to experimental preeclampsia induced by l-NAME. However, detailed data on the interrelationship between TFPI2 and HIF-1α/VEGF are lacking. Additionally, it has been reported that vascular endothelial growth factor (VEGF) can induce TFPI2 mRNA and protein expression in endothelial cells [46]. Conversely, TFPI2 inhibited VEGF-induced endothelial cell proliferation. In endothelial cells, TFPI2 and VEGF may influence each other, but this has not been studied in preeclampsia.

Therefore, the following sentence was added to the fourth paragraph of the Discussion section:

As plasma TFPI is increased and placental growth factor (PlGF) is decreased in women with preeclampsia, the TFPI-to-PlGF ratio may be useful as a predictive marker for early prediction of preeclampsia [18]. Therefore, it is of great interest to assess which marker best predict the risk for development of preeclampsia using the TFPI1- and TFPI2-specific immunoassays.

Reviewer 2 Report

Authors in this review emphasize the mechanistic role tissue factor pathway inhibitors may have in preeclampsia. While this review could be very influential for future research proposals, we do however, have some concerns that should be addressed:

1.      The authors had more discussion regarding the inclusion/exclusion criteria of the studies they included in the review, which matches more of meta-analysis type studies and gives the idea that they will re-analyze the previous data. This should be clarified.

2.      Related to literature search criteria, how was insufficient data determined (97)

3.      What do the authors mean by indisguisable TFP1 and PFP2? Many studies found changes either in TFP1 or TFP2. Hence authors must describe in detail the exclusion criteria.

4.      TFP1 has a broad role, so while discussing preeclampsia, incorporating its role in cancer appears out of context.

5.      In section 3.2.2.2 Sources. Authors neglected its expression in platelets and could have classified it in detail, while platelet activation causes the release of TFPI and influences coagulation.

6.      Section 3.2.2.3 Functional diversity… The authors added function and significance related to cancer, which seems unrelated to the review's focus.

7.      The authors did not add clinical trials in phase 1a targeting TFPI.

8.      Section 3.3. The role of TFPI1 or 2 is controversial, but summarizing the research on the role of TFPI irrespective of preeclampsia could help to understand such possible role. Many other articles need to be included, such as Res Pract Thromb Haemost. 2018 Jan; 2(1): 93–104, and similar work that are not included in any discussion. Moreover, previous work have shown that the expression of TFPi1 in platelets does not alter during preeclampsia; instead, its free plasma concentration increases. Platelets are one of the primary players in coagulation, which needs to be discussed to some extent.

9.      Current research shows consistency in TFPI data among several studies published; here author could not conclude anything new still needs to be revised.

Author Response

Answer to the reviewers

Manuscript ID: biomedicines-2322019

Type of manuscript: Review

Title: Tissue factor pathway inhibitors as potential targets for understanding the pathophysiology of preeclampsia

Authors: Hiroshi Kobayashi *, Sho Matsubara, Chiharu Yoshimoto, Hiroshi Shigetomi, Shogo Imanaka

Dear Editor in Chief:

Thank you and the reviewers for the thoughtful comments and helpful suggestions on my manuscript “Tissue factor pathway inhibitors as potential targets for understanding the pathophysiology of preeclampsia” (manuscript ID: biomedicines-2322019), authored by Hiroshi Kobayashi and colleagues. We have carefully considered each of the comments, made every effort to address the concerns raised, and applied corresponding revisions to the manuscript.

The detailed, point-by-point responses to the reviewer comments are given below, whereas the corresponding revisions are highlighted to our manuscript within the document.

We believe that our manuscript has been considerably improved as a result of these revisions, and hope that the revised manuscript is acceptable for publication in Biomedicines.

I would like to thank you once again for your consideration of our work and inviting us to submit the revised manuscript. I look forward to hearing from you.

With best regards,

Hiroshi Kobayashi, M.D., Ph.D.

Department of Gynecology and Reproductive Medicine, Ms.Clinic MayOne, Kashihara, Nara 634-0813, Japan

Department of Obstetrics and Gynecology, Nara Medical University, Kashihara, Nara 634-8522, Japan.

Tel: +81 744 29 8877

Fax: +81 744 23 6557

E-mail: hirokoba@naramed-u.ac.jp

Point-by-point responses to reviewer comments

To the Editor:

We revised the paragraphs that are highlighted in red.

Reviewer #2

Comment 1.     

The authors had more discussion regarding the inclusion/exclusion criteria of the studies they included in the review, which matches more of meta-analysis type studies and gives the idea that they will re-analyze the previous data. This should be clarified.

Response 1

As noted in response to reviewer 1, the small number of papers on TFPI2 precluded a systematic meta-analysis. Therefore, this paper is not a systematic review. This review does not follow the PRISMA (Preferred Reporting Items for Systematic Reviews and Meta-Analyses) guidelines statement. Therefore, the bias risk assessment using the Newcastle–Ottawa Scale was not performed.

Comment 2.     

Related to literature search criteria, how was insufficient data determined (97)

Response 2

We also added the following sentence in the Materials and Methods section:

Papers that did not focus on preeclampsia (e.g., inflammation, sepsis, and atherosclerosis) were excluded as insufficient data.

Comment 3.     

What do the authors mean by indisguisable TFP1 and PFP2? Many studies found changes either in TFP1 or TFP2. Hence authors must describe in detail the exclusion criteria.

Response 3

We added the following sentence in the Materials and Methods section:

Articles that could not distinguish between TFPI1 and TFPI2 were also excluded such as when only the term "TFPI" was used in the text.

Comment 4.     

TFP1 has a broad role, so while discussing preeclampsia, incorporating its role in cancer appears out of context.

Response 4

Thank you for your valuable suggestion. This is the same answer as question 6. Please see the second paragraph of the 3.2.2.3 section.

Furthermore, in addition to controlling coagulation, TFPI1 has been reported to be involved in EVT cell invasion [23]. EVT cells resemble malignant tumor cells in terms of proliferation and invasion, and the biological functions and underlying molecular mechanisms of TFPI1 in cancer cells have been extensively studied. Therefore, we first summarize the current knowledge of TFPI1 in cancer cells and then discuss its fundamental characteristics in EVT cells. In addition, although the term TFPI2 gives the impression that it acts as an inhibitor of TF, it actually exhibits prothrombotic effects in cancer cells. Due to the limited literature on the biological functions of TFPI2 in trophoblast cells, TFPI2 expression and function in cancer cells were cited for comparison.

Comment 5.     

In section 3.2.2.2 Sources. Authors neglected its expression in platelets and could have classified it in detail, while platelet activation causes the release of TFPI and influences coagulation.

Response 5

The following sentence was added.

Platelet activation actually triggers the release of TFPI1 and TFPI2. TFPI1 in platelets downregulates the initiation phase of coagulation, whereas TFPI2 not only regulates the intrinsic coagulation pathway, but also promotes fibrinolysis [39].

A literature search for the combination of “platelet activation”, “coagulation” and “tissue factor pathway inhibitor-2” yielded one hit, but it was unrelated to platelet activation [Shinoda E]. It is currently unknown whether TFPI2 released by platelet activation actually affects the coagulation process, so this point could not be fully discussed.

Shinoda E, Yui Y, Hattori R, Tanaka M, Inoue R, Aoyama T, Takimoto Y, Mitsui Y, Miyahara K, Shizuta Y, Sasayama S. Tissue factor pathway inhibitor-2 is a novel mitogen for vascular smooth muscle cells. J Biol Chem. 1999 Feb 26;274(9):5379-84. doi: 10.1074/jbc.274.9.5379.

Comment 6.      

Section 3.2.2.3 Functional diversity… The authors added function and significance related to cancer, which seems unrelated to the review's focus.

Response 6

This is the same answer as question 4. Information on the TFPI-induced proliferation, migration, invasion and apoptosis in trophoblast cells is scarce. Therefore, we first summarize information about TFPI1 and TFPI2 recently reported in cancer cells, attempting to compare it with the result in trophoblast cells.

Please see the second paragraph of the 3.2.2.3 section.

Furthermore, in addition to controlling coagulation, TFPI1 has been reported to be involved in EVT cell invasion [23]. EVT cells resemble malignant tumor cells in terms of proliferation and invasion, and the biological functions and underlying molecular mechanisms of TFPI1 in cancer cells have been extensively studied. Therefore, we first summarize the current knowledge of TFPI1 in cancer cells and then discuss its fundamental characteristics in EVT cells.

Comment 7.     

The authors did not add clinical trials in phase 1a targeting TFPI.

Response 7

We found one article showing that the IgG1 monoclonal antibody PF-06741086, which targets the Kunitz-2 domain of TFPI, has been entered in the phase II clinical trial as a candidate treatment for patients with hemophilia [Mahlangu J].

Mahlangu J, Luis Lamas J, Cristobal Morales J, Malan DR, Teeter J, Charnigo RJ, Hwang E, Arkin S. Long-term safety and efficacy of the anti-tissue factor pathway inhibitor marstacimab in participants with severe haemophilia: Phase II study results. Br J Haematol. 2023 Jan;200(2):240-248. doi: 10.1111/bjh.18495.

However, this article is aimed at treating hemophilia, not preeclampsia. A literature search was conducted combining “clinical trials”, “tissue factor pathway inhibitor”, and “preeclampsia”, but no relevant papers were found. I would appreciate it if you could point me to a related paper.

Comment 8.    

Section 3.3. The role of TFPI1 or 2 is controversial, but summarizing the research on the role of TFPI irrespective of preeclampsia could help to understand such possible role.

Many other articles need to be included, such as Res Pract Thromb Haemost. 2018 Jan; 2(1): 93–104, and similar work that are not included in any discussion.

Moreover, previous work have shown that the expression of TFPi1 in platelets does not alter during preeclampsia; instead, its free plasma concentration increases.

Platelets are one of the primary players in coagulation, which needs to be discussed to some extent.

Response 8

Gardiner et al reported that blockade of TF eliminates thrombin generation, while inhibition of TFPI increases in endogenous thrombin potential in preeclampsia [Gardiner C]. However, Egan et al. reported that plasma tissue factor pathway inhibitor (TFPI) activity was increased in early onset preeclampsia patients, and that patients with early onset preeclampsia are characterized by an attenuated coagulation response characterized by reduced thrombin generation stimulated by low-dose TF and elevated plasma TFPI activity [Egan K]. Thus, the role of TFPI is controversial. The reason is that to date there are no articles comparing total TFPI, free TFPI1, and free TFPI2 in plasma, platelets, and placental tissues of women with normal pregnancy and preeclampsia. TFPI2 in platelets from normal or pregnant subjects and in plasma from pregnant women has been reported to bind FV/Va and regulate intrinsic coagulation and fibrinolysis, as shown in ref. [39].

This was summarized in section 3.2.2.2.

It has also been pointed out that TFPI1 and TFPI2 not only have different functions, but also different biosynthetic and metabolic pathways. Ellery et al suggested that total TFPI and TFPI1 derived from plasma and platelets may be dependent on different biosynthetic and metabolic pathways [Ellery PER]. Due to the lack of other articles on platelets and TFPI2, it may be difficult to discuss further in comparison to TFPI1.

Gardiner C, Tannetta DS, Simms CA, Harrison P, Redman CW, Sargent IL. Syncytiotrophoblast microvesicles released from pre-eclampsia placentae exhibit increased tissue factor activity. PLoS One. 2011;6(10):e26313. doi: 10.1371/journal.pone.0026313.

Egan K, O'Connor H, Kevane B, Malone F, Lennon A, Al Zadjali A, Cooley S, Monteith C, Maguire P, Szklanna PB, Allen S, McCallion N, Ní Áinle F. Elevated plasma TFPI activity causes attenuated TF-dependent thrombin generation in early onset preeclampsia. Thromb Haemost. 2017 Jul 26;117(8):1549-1557. doi: 10.1160/TH16-12-0949.

Ellery PER, Hilden I, Sejling K, Loftager M, Martinez ND, Maroney SA, Mast AE. Correlates of plasma and platelet tissue factor pathway inhibitor, factor V, and Protein S. Res Pract Thromb Haemost. 2018 Jan;2(1):93-104. doi: 10.1002/rth2.12058.

Comment 9.     

Current research shows consistency in TFPI data among several studies published;

here author could not conclude anything new still needs to be revised.

Response 9

Thank you for your valuable comment.

Current research shows consistency in TFPI1 data among several studies published. Concentrations of TF and TFPI1 are often increased in maternal plasma of preeclamptic women compared with normal pregnancy, while various studies have yielded inconsistent results regarding TFPI2 [17,19,20,51,63]. In order to solve this problem, we believe that research using an assay system that can quantify TFPI1 and TFPI2 separately is necessary. This point is mentioned at the end of the discussion section.

Round 2

Reviewer 1 Report

The revised version of the paper has been significantly improved, the authors' responses to the reviewers' comments are correct. This work can therefore be published as is. Congratulations to the authors.